

# Phylogenomic analyses reveal a Gondwanan origin and repeated out of India colonizations into Asia by tarantulas (Araneae: Theraphosidae)

Saoirse Foley[1,2,3], Henrik Krehenwinkel[4], Dong-Qiang Cheng[3] and William H. Piel[2,3,5]

[1] Department of Biological Sciences, Carnegie Mellon University, Pittsburgh, PA, USA
[2] Department of Biological Science, National University of Singapore, Singapore, Singapore
[3] Division of Science, Yale-NUS College, Singapore, Singapore
[4] Department of Biogeography, Universität Trier, Trier, Germany
[5] Lee Kong Chian Natural History Museum, National University of Singapore, Singapore, Singapore

Corresponding author
Saoirse Foley, schioedtei@gmail.com

## ABSTRACT

The study of biogeography seeks taxa that share a key set of characteristics, such as timescale of diversification, dispersal ability, and ecological lability. Tarantulas are ideal organisms for studying evolution over continental-scale biogeography given their time period of diversification, their mostly long-lived sedentary lives, low dispersal rate, and their nevertheless wide circumtropical distribution. In tandem with a time-calibrated transcriptome-based phylogeny generated by PhyloBayes, we estimate the ancestral ranges of ancient tarantulas using two methods, DEC+j and BBM, in the context of their evolution. We recover two ecologically distinct tarantula lineages that evolved on the Indian Plate before it collided with Asia, emphasizing the evolutionary significance of the region, and show that both lineages diversified across Asia at different times. The most ancestral tarantulas emerge on the Americas and Africa 120 Ma–105.5 Ma. We provide support for a dual colonization of Asia by two different tarantula lineages that occur at least 20 million years apart, as well as a Gondwanan origin for the group. We determine that their current distributions are attributable to a combination of Gondwanan vicariance, continental rafting, and geographic radiation. We also discuss emergent patterns in tarantula habitat preferences through time.

## INTRODUCTION

Due to the historically close geographies of several continental land masses prior to the breakup of the supercontinents Laurasia and Gondwana, many related taxa are presently found on opposite sides of the world. For instance, it has been shown that rodents and primates (*De Oliveira, Molina & Marroig, 2009*) in the Neotropics likely arrived there from Africa. Several mass faunal exchanges between continents have occurred when two land masses eventually meet, such as the "Great American Interchange" (*Marshall et al.,*
*1982*) after the formation of the Isthmus of Panama (*O'Dea et al., 2016*), the dispersal of several taxa "out of India" and into Asia/Indomalaya (*Karanth, 2006*), and exchanges between Africa and Eurasia (*Hedges, 2001*; *Springer et al., 2011*). If we are to understand the evolutionary patterns involved in the diversification of taxa and the generation of biodiversity, it becomes important to consider the roles that continental drift may play, particularly in cases where evolutionary novelties have emerged in the diversification process.

Studies that use model taxa to understand biogeographic patterns are limited by the fact that the ideal model organism should have both a low dispersal ability and a wide (if not global) distribution. Given their low motility and wide array of dispersal strategies, plants have often been the focus of biogeographic studies (e.g., *Zhou et al., 2018*; *McGlone et al., 2018*; *Fariña et al., 2018*; *Echeverría-Londoño et al., 2018*). Mygalomorph spiders from the family Theraphosidae, also known as tarantulas, are also useful models for understanding biogeographic patterns. They possess several intriguing adaptations that have received significant attention in recent years, such as their stridulatory capacity (*Galleti-Lima & Guadanucci, 2018*; *Galleti-Lima & Guadanucci, 2019*), urticating bristle evolution (*Bertani, 2003*; *Kaderka et al., 2019*; *Foley et al., 2019*), colouration (*Foley, Saranathan & Piel, 2020*), and venom compositions (*Santana et al., 2017*; *Lüddecke, Vilcinskas & Lemke, 2019*). Robust, subfamily-level phylogenies are late in coming (*Lüddecke et al., 2018*; *Foley et al., 2019*), but these phylogenies have verified the monophylies of all studied subfamilies, except Ischnocolinae (with Selenogyrinae being the only subfamily not represented by these studies). It has only recently become possible to study tarantula adaptations in an explicitly phylogenetic context.

This diverse suite of intriguing adaptations undoubtedly contributed to the success of these creatures. They are quite widespread and are found throughout the subtropical regions of every continent (*Gallon, 2000*; World Spider *Catalog, 2020*). However, while many mature male tarantulas are known to wander in search of females (*Prentice, 1992*; *Pérez-Miles et al., 2005*; *Pérez-Miles et al., 2007*), most juvenile and female tarantulas rarely venture far from their retreats (*Yáñez & Floater, 2000*; *Schultz & Schultz, 2009*). Spiderlings seldom disperse far from the burrows of their mothers, leading to dense aggregations of burrows in relatively small areas (*Reichling, 2000*; *Shillington & McEwen, 2006*). There are no reports of ballooning tarantulas like in other mygalomorph families (*Coyle, 1983*; *Coyle et al., 1985*; *Buzzato, Haeusler & Tamang, 2021*), but some mygalomorphs are known to be highly divergent across small areas (*Starrett & Hedin, 2007*). These behaviours do not portend that tarantulas would be successful dispersers, yet they have spread across the globe and have colonized strikingly different ecological niches.

Previous studies estimate that tarantulas emerged between 150 Ma–71 Ma (*Bond et al., 2014*) or ∼107 Ma (*Opatova et al., 2020*), which is compatible with a Gondwanan origin. Indeed, some tarantulas (Selenocosmiinae) are suggested to be North Gondwanan taxa (*West, Nunn & Hogg, 2012*). Given their habitat specificity and low dispersal ability, yet wide geographic distribution, tarantulas are good model organisms for addressing biogeographic questions. In particular, two distinct lineages of Asian tarantulas were recovered in the most recent subfamily-level tarantula phylogeny (*Foley et al., 2019*). Estimates surrounding

the timing of the collision between the Indian Subcontinent and Asia vary, but most studies agree on a late Paleocene/early Eocene timescale around 55 Ma (*Hu et al., 2016*; *Zheng & Wu, 2018*; *Westerweel et al., 2019*). Migrations from the Indian Subcontinent to Asia are known to have played a significant role in generating the high faunal diversity observed in Asia today (*Karanth, 2006*; *Morley et al., 2016*; *Morley, 2018*; *Garg & Biju, 2019*), even amongst spiders (*Li, Shao & Li, 2020*) and other invertebrates (*Joshi, Karanth & Edgecombe, 2020*). It has also been shown that other spiders of a comparable age originated on the Gondwanan supercontinent, i.e., South America and Africa, before eventually colonizing Asia and Oceania (*Chamberland et al., 2018*). We expect to recover a similar pattern for tarantulas.

Here, we test the Gondwanan origin hypothesis for tarantulas using a broad sampling of molecular data and taxa. We also hypothesize that Asian tarantulas may have originated on the Indian Subcontinent and subsequently dispersed throughout Asia. We test this by inferring the historical patterns that contributed to present-day theraphosid distributions. We use a subfamily-level chronogram of Theraphosidae and estimate their ancestral ranges under two different models, allowing us to interpret the node ages in the context of geographical events. We then discuss a timeline of events that may have led to the emergence of theraphosid subfamilies, with emphasis on those found in Asia, and present some hypotheses on how those events impacted their current distributions.

# METHODS

## Data acquisition

Although theraphosids are the primary focus of this study, the poor fossil record in this group required that we expand our sampling to include other Mygalomorphae with richer fossil records so that these could assist in molecular clock calibration. Only two tarantula fossils are known, and neither can be firmly placed onto the tarantula phylogeny; one from Burmese amber dating to ~100 Mya (*Wunderlich & Müller, 2020*), and one from Chiapas amber dating to somewhere between the late Oligocene and mid Miocene (possibly as young as 16 Mya, *Dunlop, Harms & Penney, 2008*). Inferring calibration points from prior studies (i.e., secondary calibration) has also been discouraged (*Schenk, 2016*).

Transcriptome data for 29 theraphosids and 18 other mygalomorphs was obtained from the publicly available sequence read archive (SRA) database (http://www.ncbi.nlm.nih.gov/sra). Additionally, to improve the rooting of the family, we included a new transcriptome from a field-collected barychelid, *Rhianodes atratus* (collected under permit number NP/RP18-046, issued by the National Parks Board of Singapore). RNA was extracted using a TRIzol total RNA extraction protocol (*Simms, Cizdziel & Chomczynski, 1993*). The RNA was subsequently sent to a commercial company (NovogeneAIT) for paired-end sequencing on an Illumina HiSeq 4000. The resulting transcriptome is available on NCBI's Sequence Read Archive (SRA) under bioproject number PRJNA635363 (accession GIXJ00000000). Accession numbers for a total of 48 transcriptomes are included in Table S1. Assembly data for the *Rhianodes atratus* (Barychelidae) transcriptome is available in Table S2.

## A core ortholog variant to obtain a DNA supermatrix

Transcriptomes were assembled using Trinity v2.6.6 (*Grabherr et al., 2011*), and protein coding regions were predicted using TransDecoder v5.5.0 (*Haas et al., 2013*). The core-ortholog (OG) pipeline as per *Garrison et al. (2016)* was implemented for spider-specific ortholog selection across our dataset. The initial set of core orthologs was generated from a selection of publicly available transcriptomes, as per *Cheng & Piel (2018)*, and a set of 4,446 profile hidden Markov models (pHMMs) was generated following their methods. HaMStR v13.1 (*Ebersberger, Strauss & Von Haeseler, 2009*) inferred orthology between these pHMMs and our dataset. Groups of orthologous genes were then pooled together and subjected to several filtering steps.

Software used during the filtering process include the following: MAFFT (*Katoh et al., 2005*) for alignment during filtering; ALISCORE (*Misof & Misof, 2009*; *Kück et al., 2010*) for alignment trimming; ALICUT (*Kück, 2009*) for excising ambiguous regions; Infoalign (*Rice, Longden & Bleasby, 2000*) for generating consensus sequences from OGs. Generally, the filtering process and criteria was the same as in *Foley et al. (2019)* with one key exception—instead of removing any OG that was sampled for half (or fewer) of the taxa included, we chose to only retain OGs that were present in all taxa, discarding the rest. Despite this stringency, an adequate set of 743 OG alignments remained, each of which included all 48 taxa. To obtain a more neutral, granular dataset, these protein alignments were converted back to DNA alignments using localized tBLASTn searches for each taxon (*Camacho et al., 2009*) against their corresponding Trinity outputs, in tandem with a set of custom Perl scripts for parsing. Gene trees for each of the 743 alignments were estimated using RAxML (*Stamatakis, 2014*). These departures from previous filtering criteria were informed by the downstream needs of our analysis pipeline.

To exclude OGs with conflicting signals, we identified divergent gene trees using Robinson-Foulds (RF) metrics (*Robinson & Foulds, 1981*) in the R package "phytools" (*Revell, 2012*). Given that the theraphosid tree from *Foley et al. (2019)* received 100% bootstrap support at almost all nodes, we used this as the reference tree to compare with each gene tree. In order to calculate RF values, each of our 743 gene trees was pruned to match the set of taxa in the reference tree. Any OG DNA alignment whose corresponding gene tree received an RF score of $\geq 6$ was discarded, leaving 125 OGs.

Next, TranslatorX (*Abascal, Zardoya & Telford, 2010*) was used to realign the DNA sequences for the 125 OGs according to our protein alignments. Visualization of OGs in DAMBE v7 (*Xia & Xie, 2001*) showed that phylogenetically informative content was being distorted due to sequence saturation (*Xia et al., 2003*; *Philippe et al., 2011*), so every third base was subsequently removed to mitigate this effect. Lastly, FASconCAT (*Kück & Meusemann, 2010*) was used to concatenate all 125 OGs into a supermatrix, and GBLOCKS (*Castresana, 2002*) was subsequently used to trim and excise ambiguous regions. The result was a 96% complete DNA alignment of 48 taxa and 89,302 characters.

## Time calibration with PhyloBayes

BEAST v1.10.4 (*Drummond & Rambaut, 2007*) was used to generate an initial starting phylogeny for time calibration, running for 100 million generations with 20% of the

trees discarded as burnin. Divergence times were estimated in PhyloBayes v4.1 (*Lartillot, Lepage & Blanquart, 2009*) using this tree, the DNA supermatrix, and four fossils to infer calibration points. We followed *Opatova et al. (2020)* in treating the oldest mygalomorph fossil (*Rosamygale grauvogely*, *Selden & Gall, 1992*) as an Avicularioidea crown group, and use a subset of their fossil calibrations.

The first calibrated node corresponds to the Avicularioidea-Atypoidea split. The *Rosamygale grauvogely* fossil dated to 242 Million years ago (Ma), which represents the minimum bound for this node, while 323 Ma serves as the maximum bound and corresponds to the oldest age of the Bashkirian stage from which the oldest known spider fossil (*Arthrolycosa* sp.) hails (*Selden et al., 2014*; *Garwood et al., 2016*). Secondly, the split between Nemesioidina and Bemmeridae/Theraphosidae was assigned a minimum bound of 125 Ma based on the *Cretamygale chasei* fossil (*Selden, 2002*), and a maximum bound of 242 Ma based on *Rosamygale grauvogely*. Lastly, the split between Antrodiaetidae and Atypidae was assigned a minimum bound of 100 Ma based on a fossil of *Ambiortiphagus ponomarenkoi* (*Eskov & Zonstein, 1990*), with *Rosamygale grauvogely* once again providing the maximum bound of 242 Ma for this node.

Four independent PhyloBayes chains were each run for c. 24,000 cycles under an autocorrelated lognormal model with a GTR matrix, and otherwise default parameters. Several advantages of autocorrelated lognormal models over uncorrelated rates models have been demonstrated for a variety of datasets (*Lepage et al., 2007*; *Paradis, 2013*), and they have been shown to yield more plausible results in similar studies to ours (*Sharma et al., 2018*). Burn-in was adjusted to 10% to maximize the number of estimated sample size values that exceeded 200, yielding a total of 86,933 cycles across all chains. Divergence times were estimated using the "readdiv" command in PhyloBayes for each chain, and the results were summarized with TreeAnnotator. The resulting time-calibrated mygalomorph phylogeny is available in Fig. S1, and we proceeded with the tarantula subsection of this phylogeny for further analyses.

## Biogeographic analysis

The tarantulas included in this study were assigned to one or more of the following five biogeographic ranges as determined by the World Spider *Catalog (2020)*: (A) Indian Subcontinent; (B) Asia (non-Indian Indomalaya); (C) East of the Wallace Line; (D) Americas; and (E) Africa. Taxon range codings are included in Table S1. We also accounted for proximity between areas by excluding certain combinations of ancestral area (e.g., Asia plus Americas. Full list in Table S3). These areas were chosen based on the biogeographic realms first proposed by *Udvardy (1975)*, but we treat India separately to the remainder of Indomalaya given that we expect the age of theraphosids to exceed the collision of India and Asia.

The tarantula subsection (with outgroups) of our tree from PhyloBayes served as input for RASP (*Yu et al., 2015*), which estimated ancestral range distributions under two different models that were chosen via model testing (Table 1) in BioGeoBEARS (*Matzke, 2013*). Despite BAYAREALIKE+j being predicted as the most favorable model, its assumption of a static geological history (*Landis et al., 2013*) rendered it unsuitable. Hence, we ran the

**Table 1 Model testing.** Model appropriateness as determined by model testing in BioGeoBEARS. While BAYAREALIKE+j emerges as the most appropriate, its assumption of a static geologic history led us to choose DEC+j instead (as indicated by bold font). Asterisks indicate a significant difference between the appropriateness of a model with and without the +j. Hence, we also include DEC in supplementary Fig. S3.

| Model | LnL | AICc | AICc_wt | Δ −j/+j |
|---|---|---|---|---|
| DEC | −34.83 | 74.13 | 0.22 | |
| **DEC+J** | **−33.25 73.51** | | **0.3** | |
| DIVALIKE | −38.51 | 81.51 | 0.0055 | * |
| DIVALIKE+J | −34.18 | 75.35 | 0.12 | |
| BAYAREALIKE | −48.13 | 100.7 | 3.70E−07 | * |
| BAYAREALIKE+J | −33.1 | 73.21 | 0.35 | |

analysis using our second highest scoring model—the dispersal-extinction cladogenesis with jump dispersal (DEC+j) model (*Ree & Smith, 2008*). However, such "+j" models have been divisive, with some criticizing the "+j" parameter (*Ree & Sanmartín, 2018*) and others advocating for its inclusion (*McDonald-Spicer et al., 2019*; *Klaus & Matzke, 2019*). Therefore, Bayesian Binary MCMC (BBM) was added as a second method, and was ran under the JC69 model for 1 million generations across ten chains with sampling every 100 generations (*Ali et al., 2011*). As the model testing found no statistically significant difference between DEC and DEC+j (Table 1), DEC was added as a third method. RASP was also used to calculate an event matrix for node 21.

# RESULTS

## Ancestral ranges and ages of theraphosid clades

Our DEC+j (Fig. 1) and BBM (Fig. S3) results are largely in agreement, disagreeing at only at two nodes (per Fig. 1)—node 20 (which DEC+j estimates as present on both Africa and the Americas, whereas BBM estimates the Americas only), and node 28 (which DEC+j estimates as present on Asia only, whereas BBM estimates it as present on Asia and East of the Wallace line). Full range probabilities for DEC+j and BBM are given in Figs. S2 and S3 respectively. The DEC reconstruction is available in Fig. S4, and although there are some differences between this method and the others, the larger concepts discussed are not impacted.

Table 2 contains divergence time estimates as 95% HPDs for the theraphosid subsection of the mygalomorph phylogeny, while Table S4 contains those estimates for all nodes. Our earliest theraphosid node (20) is recovered 120 Ma–115.5 Ma in the Americas (Fig. 1), and is found sister to *Rhianodes atratus*, indicating that Barychelidae are sister to Theraphosidae. When considered alongside the divergence times, these estimations are consistent with a Gondwanan origin for Theraphosidae.

The Indian Plate is first recovered as an ancestral area at node 25 108 Ma–103.5 Ma. All ancestral range estimations recover two ecologically distinct Indian Plate lineages descending from node 25. The first such node (26) emerges 99 Ma–95 Ma, yielding several ground-dwelling taxa. It diverges into the Indian Thrigmopoeinae lineage, and the ancestor to Selenocosmiinae (node 27), which emerges 57.4 Ma–55 Ma. Node 27 represents the

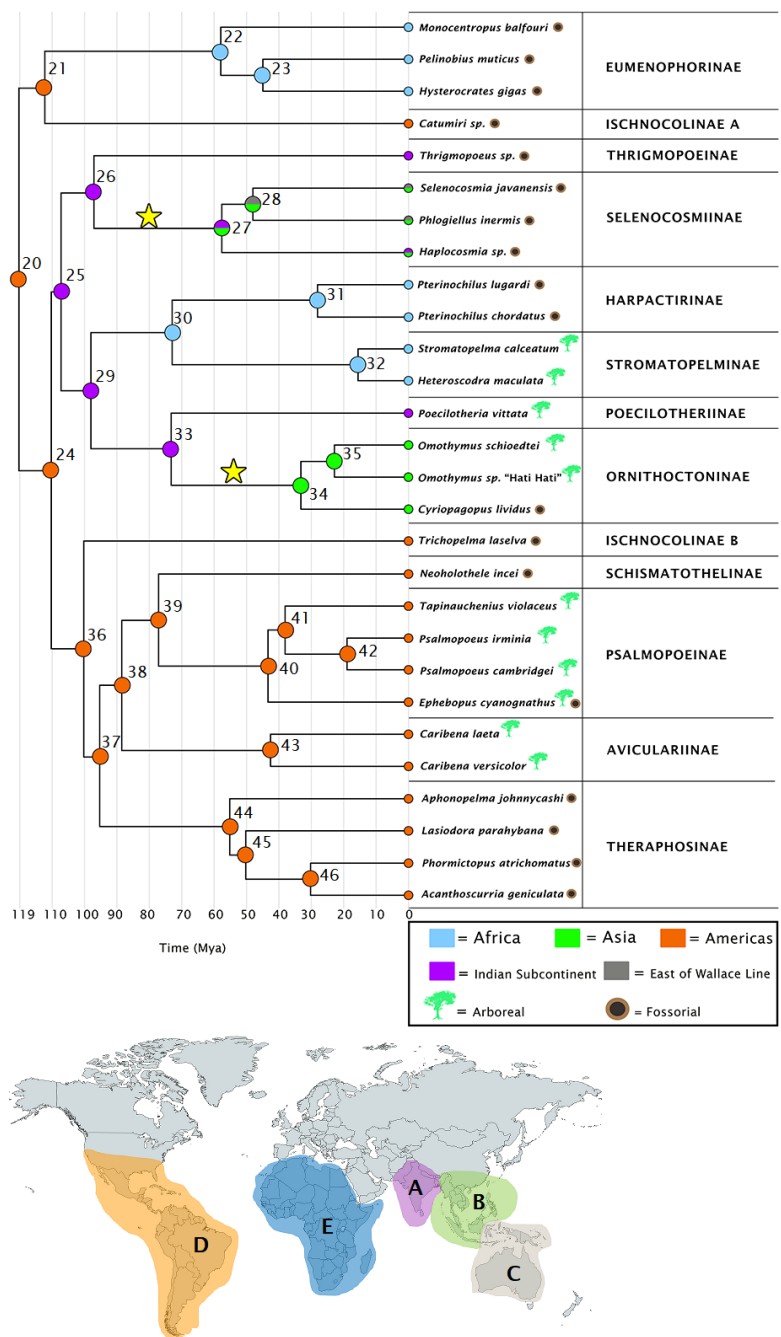

**Figure 1 Ancestral ranges as estimated under the DEC+j model.** Subfamilies are given for each species, along with a timescale in millions of years. Node numbers are provided, and each node is color coded to correspond with the areas highlighted on the map. Letters on the map correspond to range codings given in the Methods section. The stars on the tree represent independent colonizations of Asia by theraphosids, having arrived there from the Indian Subcontinent. Map credit: https://mapchart.net/, 2021. Licensed under CC BY 4.0 SA.

**Table 2 Divergence time estimates.** Divergence times with error margins per 95% HPDs as estimated by PhyloBayes. Times are given in millions of years and are rounded to the closest 0.5 Ma. Node numbers as per Fig. 1.

| Node number | Estimated age | 95% HPD Max | 95% HPD Min |
|---|---|---|---|
| 20 | 119 | 120 | 115.5 |
| 21 | 111 | 112 | 107.5 |
| 22 | 57 | 58 | 55.5 |
| 23 | 45 | 45.5 | 43.5 |
| 24 | 109 | 111 | 106.5 |
| 25 | 107 | 108 | 103.5 |
| 26 | 98 | 99 | 95 |
| 27 | 57 | 57.5 | 55 |
| 28 | 47 | 47.5 | 45.5 |
| 29 | 99 | 100 | 96 |
| 30 | 72 | 73 | 69.5 |
| 31 | 28 | 28.5 | 27 |
| 32 | 16 | 16.5 | 15 |
| 33 | 71.5 | 72 | 69 |
| 34 | 34.5 | 35 | 33 |
| 35 | 23.5 | 24 | 22 |
| 36 | 100 | 101 | 97 |
| 37 | 95 | 96 | 92.5 |
| 38 | 89 | 90 | 86.5 |
| 39 | 77 | 78 | 74.5 |
| 40 | 42 | 42.5 | 40.5 |
| 41 | 39 | 39.5 | 37.5 |
| 42 | 18.5 | 18.5 | 17.5 |
| 43 | 41.5 | 42 | 39.5 |
| 44 | 55.5 | 56 | 53.5 |
| 45 | 50.5 | 51 | 48.5 |
| 46 | 29.5 | 30 | 28.5 |

first appearance of Asia as an ancestral area in both DEC+j and BBM, albeit alongside the Indian Subcontinent (Fig. 1). This node then diverges into the *Haplocosmia* lineage, and the ancestor to *Selenocosmia* and *Phlogiellus*, which also emerges as present east of the Wallace line around 47.5 Ma-45.5 Ma (Fig. 1).

The second descendant of the first Indian Subcontinent node (25) corresponds to the last common ancestor of Harpactirinae and Ornithoctoninae (node 29), emerging 100 Ma–96 Ma. This node subsequently diverges into the ancestor of Harpactirinae and Stromatopelminae (node 30), and yet another Indian Subcontinent node corresponding to the ancestors of the primarily arboreal Ornithoctoninae, and arboreal Poecilotheriinae (node 33). After the divergence of Ornithoctoninae and Poecilotheriinae, we see Asia represented as an ancestral area for the second time in both DEC+j and BBM as the ancestor to Ornithoctoninae (node 34) emerges 35 Ma–33 Ma. DEC, however, places

**Table 3  Event matrix at node 21.** The ancestor to both Eumenophorinae and the Catumiri lineage, node 21, can provide key insights into tarantula biogeography. DEC+j and BBM favor a dispersal event influencing this node, which is more consistent with members of Eumenophorinae crossing from South America to Africa. However, DEC favors vicariance, which would attribute the present-day distribution to continental drift.

| Event | DEC+j | BBM | DEC |
|---|---|---|---|
| Dispersal | 2 | 2 | 0 |
| Vicariance | 1 | 1 | 1 |
| Extinction | 0 | 0 | 0 |

the ancestor to Poecilotheriinae and Ornithoctoninae on both the Indian Plate and Asia (Fig. S4).

The ancestor to the primarily African Eumenophorinae, and the *Catumiri* lineage (node 21) emerges in the Americas 112 Ma–107.5 Ma in all three analyses, with DEC also placing it in Africa. The event matrices of all analyses for this node are given in Table 3.

# DISCUSSION

## The importance of India and the dual colonizations of Asia

The ''out of India'' hypothesis (*McKenna, 1973*) has been accepted as a potential distribution pattern for the radiations of a diverse array of taxa (*Karanth, 2006*; *Conti et al., 2002*; *Joshi, Karanth & Edgecombe, 2020*). Our results are consistent with two independent ''out of India'' dispersals of tarantulas into Asia (Figs. 1 and 2). This ''dual colonization of Asia via India'' pattern has also been demonstrated in centipedes (*Joshi, Karanth & Edgecombe, 2020*). Both of our Asian tarantula lineages were estimated to arrive there during timescales consistent with *Li, Shao & Li (2020)*, who determined that Ochyroceratid spiders likely arrived in Asia via India 55 Ma–38 Ma, and also with the India/Asia collision timescales (*Hu et al., 2016*; *Westerweel et al., 2019*).

All three analyses place the ancestor to Selenocosmiinae (node 27) as present on the Indian Subcontinent and Asia 57.5 Ma–55 Ma (Fig. 1)—the first entry of theraphosids into Asia per DEC+j and BBM. Furthermore, both of these analyses recover the ancestor to *Selenocosmia* and *Phlogiellus* (node 28) in both Asia and east of the Wallace line, with DEC recovering the node as Asia-only. Interestingly, despite our analyses suggesting a Gondwanan origin for Theraphosidae, this pattern suggests that tarantulas were not always present in Oceania, and instead is consistent with Selenocosmiinae having diversified across Asia, eventually crossing the Wallace line sometime after the India/Asia collision (possibly as early as 47 Ma, Fig. S3), while the terrestrial Thrigmopoeinae remained in India (Fig. 2). The second colonization event of Asia by theraphosids occurred with the common ancestors to Ornithoctoninae (node 34, Fig. 1), which emerged in Asia at least 20 million years later (35 Ma–33 Ma), while Poecilotheriinae remained in India (Fig. 2). DEC suggests that this lineage colonized Asia even earlier, placing the ancestors to Poecilotheriinae (node 33) on both Asia and the Indian Plate 72 Ma–69 Ma, but this seems too early to be plausible as (i) it predates the collision of the Indian Plate into the rest of Asia, and (ii) no extant

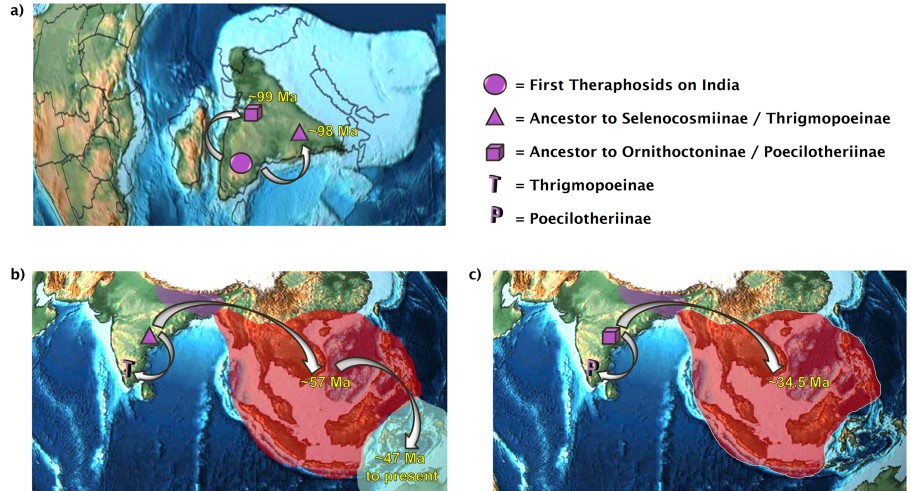

**Figure 2 The dual colonization of Asia.** Two independent tarantula lineages emerge out of India. (A) The first theraphosids on the Indian Plate diverge into the ancestor of Selenocosmiinae and Thrigmopoeinae (98 Ma) and into the ancestor of Ornithoctoninae and Poecilotheriinae (99 Ma); (B) present-day distributions of Selenocosmiinae. They entered Asia via India 57 Ma and crossed the Wallace line sometime after 47 Ma. Some Selenocosmiinae (e.g., Haplocosmia nepalensis) may also be found on the Indian Subcontinent today, where Thrigmopoeinae remain; (C) present-day distributions of Ornithoctoniinae, which entered Asia via India 34.5 Ma but did not manage to cross the Wallace line. Meanwhile, Poecilotheriinae remain on the Indian Subcontinent. Map credit: https://dinosaurpictures.org/, 2021. Licensed under CC BY 4.0 SA.

members of Poecilotheriinae exist in the rest of Asia, and DEC did not predict an extinction event at this node.

Our results indicate that both of these Asian lineages diverged while the Indian Plate was still rafting towards Asia, as the collision would likely not have happened before node 25 diverges to yield nodes 26 and 29 (Table 2), both of which also emerge on the Indian Plate in our DEC+j and BBM analyses (Fig. 1 and Fig. S3), while DEC also places node 29 on the African continent in addition to the Indian Plate. Interestingly, the two lineages also appear to be ecologically divergent. Selenocosmiinae is a subfamily that almost exclusively contains terrestrial members, whereas Ornithoctoninae is a largely arboreal subfamily with terrestrial members.

A theraphosid fossil was recently uncovered in Burmese amber, dating to ∼100 Ma (*Wunderlich & Müller, 2020*). While the identity of the fossil cannot adequately be established (though the authors did speculate that it may represent an ancestor to Selenocosmiinae), Burmese amber is thought to be of Gondwanan origin (*Oliveira et al., 2016*; *Poinar Jr, 2018*). The location and age of this fossil could plausibly correspond to one of the ancestral tarantula groups at either node 21 or 24, and confers some support towards the validity of our ancestral area estimations and dating.
## The Catumiri lineage as a biogeographic relic, and a journey to Africa for Eumenophorinae

Our estimate for the age of our most ancestral theraphosid node (20) was 120 Ma–105.5 Ma (Table 2), which is consistent with previous estimates based on fewer ingroups that dated the divergence of tarantulas to occur 150 Ma–71 Ma (*Bond et al., 2014*) or ∼107 Ma (*Opatova et al., 2020*). All three analyses recover this node in the Americas, but DEC also places it on Africa. South America finally split from Africa 105 Ma–100 Ma (*Heine, Zoethout & Müller, 2013*; *Matthews et al., 2016*; *Olyphant, Johnson & Hughes, 2017*). While oceanic crossings were still required, it has been suggested that dispersal between both continents may have been facilitated via the Rio Grande Rise and Walvis Ridge up until ∼40 Ma, as these might have shortened the terrestrial distance between both continents (De Oliveira, Molina & Marroig, 2009).

The ancestor to the primarily African Eumenophorinae and the *Catumiri* lineage (node 21, Fig. 1) is recovered by all analyses in the Americas 112 Ma–107.5 Ma (with DEC also placing it in Africa)—a period when South America and Africa were likely still connected. As a member of the taxonomically troubled Ischnocolinae, *Catumiri* and its relatives have been considered to be among the most basal theraphosids (*Raven, 1985*; *Schmidt, 2003*; *Guadanucci, 2014*). The findings from *Opatova et al. (2020)* and *Foley et al. (2019)* seem to agree with this with respect to *Catumiri*, which is recovered as an early-branching theraphosid in both works. *Catumiri* was also the only neotropical taxon from *Foley et al. (2019)* that was not recovered in an otherwise monophyletic group of neotropical theraphosids. With a potential maximum age of just under 107.5 Ma (Table 2), we suggest that the *Catumiri* lineage represents an ancient biogeographical relic that diversified and remained in South America while the continents drifted apart—potentially even before South America split from Africa. This could also plausibly explain why Ischnocolinae (the subfamily in which *Catumiri* is included) was the only paraphyletic subfamily recovered by *Foley et al. (2019)*.

Meanwhile, the ancestor to Eumenophorinae emerges solely in Africa in all three analyses 58 Ma–55.5 Ma. This result leaves two potential biogeographic origins for Eumenophorinae: (i) a vicariant event caused by the split between South America and Africa, which resulted in them being separated from the *Catumiri* lineage; or (ii) a dispersal event, where some of the ancestors of the *Catumiri* lineage crossed from South America to Africa and subsequently diversified. Per the event matrix in Table 3, the first scenario is supported by DEC (Fig. S4), which recovers the ancestors to both groups on both South America and Africa, whereas DEC+j and BBM favor a dispersal event, which is more consistent with a scenario where Eumenophorinae crossed back to Africa from South America, which may still have been possible as per the results from De Oliveira, Molina, and Marroig (2009).

A second emergence of tarantulas on Africa occurs at the ancestor to Harpactirinae and Stromatopelminae 73 Ma–69.5 Ma (node 30, Fig. 1), having apparently crossed back via the Indian Plate. The upward bound of this arrival can be attributed to the minimum age of its parent node (29), which is estimated at 96 Ma (Table 2), and a minimum of 69.5 Ma (minimum age of the ancestor to Harpactirinae and Stromatopelminae). Crossing between Africa and the Indian Plate via Madagascar could have been possible up until around 90

Ma to 85 Ma (*Ali & Aitchison, 2008*), and our estimates hence afford a short window for this node to cross to Africa. There are a handful of extant Malagasy theraphosids (World Spider *Catalog, 2020*) that could provide further support for this link, though sample material from these taxa was not available. This "back to Africa" event would represent an intriguing departure from typical patterns associated with Gondwanan vicariance.

### Tarantula radiations and lifestyle switches

There are three nodes in particular (20, 35, and 37, Fig. 1) where the two descendent lineages demonstrate high fidelity to habitat choices. For node 20, one descendant lineage corresponds to the fossorial Selenocosmiinae and Thrigmopoeinae, while the other corresponds to the arboreal Poecilotheriinae and Ornithoctoninae (except *Haplopelma*, which is fossorial). Node 35 splits into Harpactirinae (fossorial) and Stromatopelminae (arboreal). Node 37 splits into Theraphosinae (fossorial), and Aviculariinae, Psalmopoeinae (both arboreal) and Schismatothelinae (fossorial). These habitat choices are maintained throughout radiations across large areas, indicating that, based on our phylogeny and sampling, tarantula lineages may be demonstrating niche conservatism in showing high fidelity to lifestyle and habitat choices.

This idea is also relevant to our two independent lineages of tarantulas that colonized Asia. After the terrestrial Selenocosmiinae diversified across Asia, there may have been an opportunity for Ornithoctoninae to capitalise on an open arboreal niche, and subsequently colonize Asia and further diversify while largely retaining this newly acquired arboreal preference. The two lineages that did not colonize Asia/remained on the Indian Plate are also ecologically divergent, with Poecilotheriinae being an arboreal subfamily and Thrigmopoeinae occupying terrestrial niches, suggesting that their lifestyle differences may have facilitated their co-diversification throughout the Indian Subcontinent. The divergence times imply that the arboreal Poecilotheriinae could be as old as 69 Ma, and the terrestrial Thrigmopoeinae could be as old as 95 Ma (Table 2), which implies that this co-diversification might have occurred before the Indian Plate fully collided with Asia.

Perhaps these radiations can be attributed to an ancient switch in lifestyle that each ecologically distinct subfamily to become successful by exploiting different ecological niches. Although the subfamilies in the phylogeny used here are not represented by all members, we expect that the addition of further taxa would verify this idea, given that the niches of most higher theraphosid taxa appear to be conserved over large timescales, and that lifestyle changes are observed at the base of many theraphosid radiations. With the advent of further data from more representative taxa, we encourage future studies to test whether these lifestyle switches directly precede evolutionary rate shifts.

## CONCLUSIONS

The Indian Plate played a pivotal role in the history of Theraphosidae and facilitated the colonization of Asia by two ecologically distinct tarantula lineages at least 20 million years apart. Ancient tarantulas appear to have undergone several diversifications on India while it was still rafting, affirming the evolutionary significance of the subcontinent. We show that tarantulas demonstrate pronounced niche conservation, which could explain the ecological

divergences observed in the two independent Indian subcontinent lineages that colonized Asia. We propose a Gondwanan origin for Theraphosidae, although their present-day distributions suggest that continental rafting and geographic radiations facilitated their colonizations of subsequent landmasses.

## ACKNOWLEDGEMENTS

Thanks are due to Danwei Huang and Killian O' Connor for their valuable discussions during this project. We are grateful to David Court for his assistance with specimen collecting. Dinosaurpictures.org kindly allowed us to use their "Ancient Earth" maps to make Fig. 2.

### Funding

The computational results reported in this study were supported by a Yale-NUS College Shared Instrumentation Grant (IG15-SI101) awarded to William H. Piel. William H. Piel was also supported by the South East Asian Biodiversity Genomics (SEABIG) centre, the Singapore Ministry of Education (grant number MOE2016-T2-2-137), and by Yale-NUS College (grant numbers IG14-SI002 and R-607-265-052-121). The funders had no role in study design, data collection and analysis, decision to publish, or preparation of the manuscript.

### Grant Disclosures

The following grant information was disclosed by the authors:
Yale-NUS College Shared Instrumentation: IG15-SI101.
South East Asian Biodiversity Genomics (SEABIG) centre.
Singapore Ministry of Education: MOE2016-T2-2-137.
Yale-NUS College: IG14-SI002, R-607-265-052-121.

### Competing Interests

The authors declare there are no competing interests.

### Author Contributions

- Saoirse Foley conceived and designed the experiments, performed the experiments, analyzed the data, prepared figures and/or tables, authored or reviewed drafts of the paper, and approved the final draft.
- Henrik Krehenwinkel, Dong-Qiang Cheng and William H. Piel conceived and designed the experiments, analyzed the data, authored or reviewed drafts of the paper, and approved the final draft.

### Field Study Permissions

The following information was supplied relating to field study approvals (i.e., approving body and any reference numbers):

The National Parks Board of Singapore granted permission to collect Rhianodes atratus (Permit number NP/RP18-046).

## DNA Deposition

The following information was supplied regarding the deposition of DNA sequences:

The resulting transcriptome is available on NCBI's Sequence Read Archive (SRA): PRJNA635363, GIXJ00000000.

Accession numbers for a total of 48 transcriptomes and assembly data for the Rhianodes atratus (Barychelidae) transcriptome is available in the Supplemental Files.

## Data Availability

Data are available in the Supplemental Files.

## Supplemental Information

Supplemental information for this article can be found online at http://dx.doi.org/10.7717/peerj.11162#supplemental-information.

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
