# Peer review of "Phylogenomic analyses reveal a Gondwanan origin and repeated out of India colonizations into Asia by tarantulas (Araneae: Theraphosidae)"

_PeerJ, doi:10.7717/peerj.11162_

## Round 0.1 · original submission · Major Revisions

I have received now three reviews about your manuscript recently submitted to PeerJ. All of them were very positive about the paper, while also pointing out aspects that deserve to be improved. R2 also mentioned that the error bars in the node dating are incredibly small. Please, provide an explanation for that.

·

Basic reporting

The manuscript entitled “Phylogenomic analyses reveal a Gondwanan origin and repeated out of India colonizations into Asia by tarantulas (Araneae: Theraphosidae)” is clear and concise providing an easy reading. The introduction presents a clear background of the topic and the bibliography is well referenced and relevant. Figures and supplementary material are accurate and clear.

**One minor detail is in the supplementary material (Supp1_DEC+j_Probs and Supp2_BBMProbs) the different tones of red and blue on the square references can´t be distinguish in the nodes (circle references).

Experimental design

Despite that mygalomorph´s spiders are excellent models for biogeographic analyses, studies on this spider infraorder are scarce. This manuscript presents new insights on the evolution on the distribution of this group. The methodology used is suitable and concordant with other studies published. It is well described and well referenced.

Validity of the findings

Results obtained are robust and well represented in figures/tables. Discussion is well justified and is concordant with the results and previous studies. The authors propose a Gondwanan origin of Theraphosidae and a dual colonization from India to Asia. This manuscript contributes to the knowledge of the Mygalomorphae evolution providing new hypothesis for their current distribution (vicariance, continental rafting, geographic radiation).

Additional comments

Very nice manuscript.

·

Basic reporting

This is a nice study, whose findings do support their ideas.

#I was a reviewer on a previous version of this manuscript. I feel they have addressed most of my previous recommendations with regards to methodolgies (especially with regards to the available fossils for tarantulas and the use of secondary calibrations).
But I do have a few issues, once is critical (either needs to be explained or should be reevaluated due to potential issues).

See below for my explanations

Experimental design

no comment

Validity of the findings

no comment

Additional comments

I am confused by the # of OGs. You had 743 and then say you had 125 after the RF metrics... you lost that many loci??? That seems really problematic.
Also, using the tree you previously generated (by using the same loci/type of dataset) seems circular and just produces potential bias.

not addressed from my previous review of this manuscript:
- why convert amino acids back to nucleotides and then use that for phylogeny inference? At this level of inference/investigation, why not just use amino acids?
- why start in BEAST, then switch to PhyloBayes?

line 169 - italicize Rosamygale grauvogely

Results
*SERIOUS* - The error bars in the dating analysis are incredibly small (tight)...something I don't feel like I've ever seen before, especially for a deep-time analysis like this across a divergent group of organisms. It just seems like something is wrong. Was this investigated?

The results are consistent with the “out of India” hypothesis, but why are you not discussing this VERY important radiation back into Africa (the Harpactirinae and the Stromatopelminae), from the Indian subcontinent?

"Tarantula radiations seem to occur with lifestyle switches" - I would say that this is a bit tough to say...given the sampling. This could definitely change if more lineages are included. I would be careful with the wording here.

I would also say that "habitat choices are maintained throughout radiations across large areas" is the equivalent of niche conservatism, no?

Reviewer 3 ·

Basic reporting

no comment

Experimental design

no comment

Validity of the findings

no comment

Additional comments

This is an important contribution to the understanding of the systematics and biogeography of Theraphosidae spiders. Although not many new sequences are generated, the authors compile a meaningful number of sequences from online repositories and reanalyze them under the lens of new question, such as are theraphosids of Gondwanan origin? What is the origin of Asian Theraphosidae?

All sections of the manuscript (intro, methods, results and discussion) are well written, but several minor issues should be addressed before publication, a list of which can be found below:

Line 109: correct “dating to dating”

Lines 149-150: “Given that the theraphosid tree from Foley et al. (2019) received 100% bootstrap support at almost all nodes, we used this as the reference tree to compare with each gene tree.” Phylogenomic and transcriptomic analysis usually get 100% bootstrap support due to the massive amount of data that always support a given topology. One way to assess if the highly supported branches are found across the genes is to run a concordance factor analysis (performed by IQTREE). It is quite simple, and you can use your data in the format as is: http://www.robertlanfear.com/blog/files/concordance_factors.html

Line 169: “Rosamygale grauvogely” should be in italics.

Line 195: “(A) Indian 195 Subcontinent; (B) Asia; (C) East of the Wallace Line; (D) Americas; and (E) Africa”
These biogeographic areas were delimited based on which criteria? Did you use a reference from literature? There is plenty of references showing the importance of delimiting reasonable biogeographic areas, e.g. https://www.researchgate.net/publication/322820908_Biogeographical_units_matter
The biogeographic areas indicated as ABCDE are used only in the Supplementary Table 1 and supplementary figures. Why not use it in the text and in Figure 3? It will surely be helpful and will make the results more intelligible. While at it, the Supplemental Table 1 could also include a column with a more precise location for the species (at least at country level), not only the geographic range.

Line 201: “via model testing (Table 1) BioGeoBEARS”
There is something missing here, it should probably be “in BioGeoBEARS”

Lines 205-206: Although the “j” parameter have been criticized recently, it has also received support and further detailing in its use: https://onlinelibrary.wiley.com/doi/abs/10.1111/jbi.13496; https://academic.oup.com/sysbio/article/69/1/61/5490843

Line 213: Ages of theraphosid clades
The first paragraph of this section contains a mix of information. The first sentence is about the phylogenetic position of Rhianodes atratus (which is mostly expected), and the rest of the paragraph is about incongruences between the biogeographic analyses, but no dates are mentioned at all. I suggest this paragraph to be moved somewhere else in the text.

Line 224: “is recovered between 120Ma and 115.5Ma in the Americas (Figure 3).”
Figure 3 doesn’t show the variance bar, that is Figure 2.

Line 227: “The Indian Plate is first represented between 108Ma and 103.5Ma (node 25).” The Indian Plate is not represented in your dating analysis, the age of the node is what is being recovered in agreement with the position of the Indian Plate. Please rephrase.

Line 227: Here the text gets a bit confusing because the node numbers are shown in Figure 2, while the image with a better resolution of species names and with the ancestral areas reconstructed in nodes are in Figure 3. So, when you say “The first such node (26) emerges between 99Ma and 95Ma” I can see in figure 2 that node 26 in over that date range, but I don’t know which ancestral area it occupies, so I move to figure 3, but there’s no node 26 in figure 3. So, I have to go back to figure 2, lean in to try to read the species names, go back to figure 3, locate the clade and finally visualize what the biogeography of the group is. This kind of back and forth of figures happens during the whole results and discussion. Maybe this could be simplified by adding the node numbers in Figure 3.

Lines 227-235 (and in several parts of the text): “between 108Ma and 103.5Ma; between 99Ma and 95Ma; between 57.4Ma and 55Ma; between 47.5Ma and 45.5Ma”.
All this can be written in a more succinct way with a range, such as 108¬–103.5 Myr, 99–95 Myr, 57.4¬–55 Myr …

Line 248: “The event matrices of all analyses for this node are given in Table 2.” This should be in the methods section.

Line 261, 262: Please, add node numbers to Figure 3, it will make the life of readers much easier. Another option is, when citing Figure 3, not referencing node numbers at all, because we can already guide ourselves with species and clade names.

Line 311: “which split off quite close to the root of the tree”
Catumiri does not split off close to the root of the tree (it does only in figure 3, which is a crop of the larger phylogeny, which includes the actual root of the tree).

Figure 1: This figure doesn’t have much information (like distribution points) so there’s no reason for it to be this big. I suggest placing it, reduced in size, together with Figure 3, this way it will easier for the reader to associate the colors of the biogeographic regions with the colors on the tree.

Figure 2: the years in the time scale are negative and should be corrected to non-negative values.

Figure 3: Maybe change the star to another symbol to represent independent colonization of Asia; you already use starts to represent calibration nodes in Figure 2 and it gets confusing as readers reach figure 3 with a preconception of the meaning of a star in a dated phylogeny.

All supplementary tables should have legends!

Supplementary table 4 contains information that are too important to be buried into supplementary materials. It should be reformatted and included in the main paper (at least the theraphosid part of it).

---

## Round 0.2 · accepted · Accept

Thank you for resubmitting your paper. I'm satisfied by all the answers you provided to the reviewers' previous suggestions and I'm glad to accept the manuscript for publication as is.